# Community Pharmacist-Administered COVID-19 Vaccinations: A Pilot Customer Survey on Satisfaction and Motivation to Get Vaccinated

**DOI:** 10.3390/vaccines9111320

**Published:** 2021-11-14

**Authors:** Dominik Stämpfli, Adrian Martinez-De la Torre, Elodie Simi, Sophie Du Pasquier, Jérôme Berger, Andrea M. Burden

**Affiliations:** 1Pharmacoepidemiology, Institute of Pharmaceutical Sciences, Department of Chemistry and Applied Biosciences, ETH Zurich, CH-8093 Zurich, Switzerland; adrian.martinez@pharma.ethz.ch (A.M.-D.l.T.); andrea.burden@pharma.ethz.ch (A.M.B.); 2Community Pharmacy, Centre for Primary Care and Public Health (Unisanté), University of Lausanne, CH-1011 Lausanne, Switzerland; elodie.simi@unisante.ch (E.S.); sophie.du-pasquier@unisante.ch (S.D.P.); jerome.berger@unisante.ch (J.B.); 3Center for Research and Innovation in Clinical Pharmaceutical Sciences, Lausanne University Hospital, University of Lausanne, CH-1011 Lausanne, Switzerland; 4School of Pharmaceutical Sciences, University of Geneva, CH-1206 Genève, Switzerland; 5Institute of Pharmaceutical Sciences of Western Switzerland, University of Lausanne, CH-1011 Lausanne, Switzerland

**Keywords:** SARS-CoV-2, COVID-19, vaccination, satisfaction, survey, community pharmacy

## Abstract

In response to the coronavirus disease 2019 (COVID-19) pandemic, Swiss health authorities approved and ordered two mRNA vaccines in 2021. The canton of Zurich was the second in Switzerland to allow community pharmacists to administer the COVID-19 mRNA Vaccine Moderna to the adult population. We aimed to pilot a customer satisfaction questionnaire regarding COVID-19 vaccinations in Zurich pharmacies. Questions focused on satisfaction with different aspects of the service, motivation for getting the vaccination, and reasons for being vaccinated in a pharmacy. Zurich pharmacies administered 68,169 COVID-19 doses until June 2021, and 421 questionnaires were filled. Respondents’ mean age was 43.5 (±13.2) years, with 42.3% reporting being women and 46.1% being men. Of the 372 complete questionnaires, 98.7% of the respondents would have recommended the service to others. High levels of satisfaction were reported concerning pre-vaccination discussion (98.9%), pharmacies’ information level on COVID-19 vaccines (98.9%), general comfort with receiving the vaccination in a pharmacy (99.5%), injection technique (99.2%), and premises used (98.1%). Most respondents (57.3%) would have had the option of another vaccination provider, but the pharmacies were chosen for their opening hours, ease of access, and perceived trust. The availability of pharmacist-administered services may be an important contributor to a successful vaccination programme in Switzerland.

## 1. Introduction

The first case of a coronavirus disease 2019 (COVID-19) patient in Switzerland was detected on the 24 February 2020. Since then, more than 700,000 cases have been detected, leading to around 30,000 hospitalisations and 1100 deaths [1]. At the end of January 2021, two vaccines received approval by the Swiss Agency for Therapeutic Products (Swissmedic) and were made available to the Swiss population (BNT162b2 from Pfizer/BioNTech, “Comirnaty COVID-19 vaccine”, and mRNA-1273 from Moderna, “COVID-19 mRNA Vaccine Moderna”) [2,3]. Swiss authorities defined a public health strategy prioritising vaccine access for COVID-19 risk groups, and defined vaccine delivery rates [3]: the vaccines were first administered to the population at risk of severe infection, then to healthcare professionals; next, caregivers and adults in community facilities with an increased risk of infection and outbreaks; finally, the vaccination was made accessible to the general adult population, and highly encouraged to lift everyday life restrictions. Vaccination was always administered voluntarily in every population [3,4].

Since 2015, Swiss community pharmacists may be accredited for vaccinating healthy adults [5,6], and cantonal health authorities define which vaccinations are allowed to be administered in community pharmacies, and may state additional requirements and limitations [7]. Among the 26 Swiss cantons, the canton of Valais was the first to provide the COVID-19 mRNA Vaccine Moderna in community pharmacies at the end of April 2021 [8], closely followed by the canton of Zurich on the 5 May 2021 [9]. Pharmacies were eligible to administer the COVID-19 vaccines if they were in accordance with cantonal regulations and strategy [10]. As of June 2021, pharmacies performed immunization on appointment only, to provide satisfying conditions for pre-vaccination discussions, a 15-min post-vaccination observation period, and to ensure secure customer flows in the pharmacy.

Previous studies on vaccination in Swiss pharmacies considered well-known vaccines, such as influenza, and showed high levels of satisfaction and general comfort [11]. This is different for COVID-19, where mRNA vaccines are available and vaccination centres can provide the vaccination. The early rollout of the COVID-19 vaccination programme in community pharmacies of the Swiss canton of Zurich provided an opportunity to assess the service and to conduct a pilot study for a national survey. We aimed to (1) evaluate customer satisfaction with COVID-19 vaccinations in Zurich public pharmacies and (2) gain insights into the motivation to get vaccinated and for choosing a pharmacy as the vaccination site with a pilot questionnaire.

## 2. Materials and Methods

### 2.1. Study Design

We collected survey data from voluntarily participating pharmacies from the canton Zurich during vaccination roll-out from 10 May to 26 June 2021. Pharmacies were asked via the local pharmacists’ association (“Apothekerverband des Kantons Zürich”, AVKZ, CH-8037 Zurich, Switzerland) to provide vaccinated customers with an invitation letter. The invitation letter included information on the survey, contact information, and a link and scannable QR code to the online questionnaire. The customers were asked to fill out the survey during the post-vaccination observation period in the pharmacy or once back at home. Participation in the survey was voluntary for the customers as well. The survey did not collect any information related to the identity of the customer or the pharmacy.

### 2.2. Survey

The pilot survey consisted of an online questionnaire in German language (Appendix A), designed using SelectSurvey.NET version 5 (ClassApps, Kansas City, MO 64108 USA). The questionnaire was adapted from a previous study on pharmacist-administered influenza vaccinations [11] and included four main topics: (1) satisfaction with the service (feeling comfortable, injection technique, pre-vaccination discussion, pharmacy staff knowledge about the vaccines, premises/facility, recommending the pharmacy COVID-19 vaccination service to others, used vaccine, scheduling); (2) motivation for getting the COVID-19 vaccination (reducing the risk for oneself and others, increased risk due to occupation for oneself or others, restrictions in daily life, travel restrictions, following the official recommendations); (3) reason for getting the vaccination in a pharmacy (only place available, opening hours, accessibility, positive previous experiences, availability of a primary care physician, trust in the pharmacy, if they would have gotten the vaccination elsewhere as well); and (4) demographic information (sex as salutation, age, highest level of education, belonging to a COVID-19 risk group, which vaccine they had received, their general vaccination coverage). Additionally, the survey asked about the willingness for getting a booster shot (if necessary), for getting this booster shot in a pharmacy, if the customers had done an antibody test before deciding on getting the vaccination, how many vaccinations the customers had previously received in a pharmacy, and how many influenza vaccinations the customers had received in previous years. Wherever appropriate, a Likert scale with six plus one labelled levels was used, ranging from “does not apply at all” to “does fully apply” with a “not answerable” option. Respondents were free not to answer questions.

### 2.3. Statistical Analysis

We conducted a descriptive analysis of the aggregated survey data. Demographic information was stratified by sex, and summarised using means and standard deviations, or counts and proportions, as appropriate. Significant differences across groups were tested using chi-squared and *t* tests, as appropriate. Additionally, we tested for significant differences between men and women excluding the blank (i.e., missing information) sex group. For the Likert scales, counts and proportions were used, omitting “not answerable” responses and questionnaires with missing demographic information. As a sensitivity analysis, we included those respondents who did not provide demographic information. Filled questionnaires without informed consent were excluded from the analyses. All analyses were performed in R [12] with the additional packages gglot2 [13], dplyr [14], and data.table [15].

## 3. Results

### 3.1. Demographics

There were 421 filled questionnaires at the end of the data collection period. Table 1 provides the demographic information of the respondents. The mean age was 43.5 (±13.2) years. Sex was reported to be women in 42.3% and men in 46.1% of respondents, and was left blank in 49 (11.6%) returned questionnaires. No significant differences were found when comparing men versus women excluding the 49 respondents that did not report sex. Self-reported highest acquired level of education was mainly higher education (47.7%), i.e., universities and universities of applied sciences. Respondents were mainly not part of any defined COVID-19 risk group (68.4%). Respondents classified themselves as persons with a high vaccination coverage, following the recommended basic and additional vaccination programme (54.6%); but only 29.8% reported having received an influenza vaccination in recent years. For 72.9% of the respondents, this was the first vaccination in a pharmacy. COVID-19 mRNA Vaccine Moderna was the most frequent self-reported vaccine (87.4%). It was stated by 7.1% of the respondents that they had an antibody test done before deciding on the vaccination.

### 3.2. Choice of a Pharmacy as Place of Vaccination

Questionnaire items on the reasons for choosing a pharmacy as the place of vaccination are visualised in Figure 1, and numbers are reported in Appendix A. In 372 questionnaires with complete demographic information, over half of the respondents indicated that pharmacies were chosen despite having other options (57.3%). Opening hours and ease of access (e.g., parking, public transport) were cited by the majority as decisive reasons (66.8%; 85.6%, respectively). Trust in pharmacies was reported by 96.0% of respondents, although this was not usually based on positive previous experiences with vaccinations in pharmacies (57.1%). The absence (74.5%) or unavailability (65.3%) of family doctors did not lead to the choice of the pharmacy as a vaccination centre.

### 3.3. Satisfaction

Questionnaire items on satisfaction with the service are visualised in Figure 2, and numbers are reported in Appendix A. In 372 questionnaires with complete demographic information, satisfaction with the service was very high: 367 of the 372 respondents (98.7%) would have recommended it to others. Satisfaction included all the elements queried: pre-vaccination discussion (98.9%), general comfort with COVID-19 vaccination in a pharmacy (99.5%), injection technique (99.2%), and premises used (98.1%). Pharmacies were rated as very well (63.8%) or well (27.3%) informed regarding COVID-19 vaccines. Only 0.8% of participants were not satisfied with the vaccine available (COVID-19 mRNA Vaccine Moderna). There were some complaints about the appointment procedure (9.1%).

### 3.4. Motivation

Questionnaire items on the motivation to get vaccinated are visualised in Figure 3, and numbers are reported in Appendix A. In 372 questionnaires with complete demographic information, most of the reasons given for vaccination were to reduce one’s own risk (93.5%) and others’ risk (97.6%). Fewer restrictions in everyday life (97.3%), being able to travel again (93.3%), and following the recommendations of the Federal Office of Public Health (90.2%) were also largely affirmed as reasons for getting the vaccine. Occupational risks, either their own risk (45.5%) or concerns that their occupation may increase the risk for others (27.5%), were cited less frequently as a reason in this population. Additionally, 87.3% of respondents replied positively to “I would be willing to do an annual booster (if needed)”, and 95.9% of the respondents agreed to getting a booster shot in a pharmacy, with 50.1% strongly agreeing.

## 4. Discussion

This report responds to a call for studies investigating patient acceptance and experiences of pharmacists’ involvement in the COVID-19 vaccination process [16], and is the first to describe customer satisfaction, motivation to get vaccinated, and for choosing a pharmacy as a vaccination site in the context of COVID vaccinations in public pharmacies. During the roll-out of the COVID-19 vaccination programme in public pharmacies of the Swiss canton of Zurich, we identified very high satisfaction with pharmacist-administered COVID-19 vaccinations. Results from our voluntary pilot questionnaire showed that 367 of the 372 respondents with full demographic information would recommend it to others and would come back for a booster shot if necessary.

The high levels of satisfaction and general comfort (i.e., feeling comfortable) are comparable to results concerning influenza vaccinations [11,17,18,19]. During the 2019/2020 influenza season, 99% of customers (649 of 651) similarly felt comfortable with Swiss pharmacist-administered influenza vaccinations [11]. As for influenza vaccinations, satisfaction also included the element pre-vaccination discussion. This was combined with satisfaction with the pharmacies’ information level on COVID-19 vaccines. Pre-vaccination discussions and knowledge about the vaccines may be accredited to the extensive vaccination accreditation process for pharmacists (2.5 days of training followed by biennial refresh courses) and the pharmacists’ previous experience of conducting pre-vaccination discussions. In addition, various private course providers were approached by the local and national pharmacists’ associations to offer mandatory online lectures, practical workshops, and follow-up webinars specifically for COVID-vaccinating pharmacy staff. These efforts may contribute to the dissemination of quality information on COVID-19 vaccines in the population and help uptake of the COVID-19 vaccination.

Convenience because of fitting opening hours and ease of access was a primary driver for selecting a pharmacy as a place of COVID-19 vaccination, which is comparable to pharmacist-administered influenza vaccinations in Switzerland [11] and Canada [17,18]. Respondents were mainly (68.4%) not part of any defined COVID-19 risk group and belonged to a young age group (43.5 ± 13.2 years). Their motivation primarily stemmed from a general risk reduction and the prospective possibility to lift personal restrictions. These results indirectly show the benefits of including public pharmacies in vaccination programmes: they can assist in overcoming structural barriers to vaccination [20]. Pharmacies have the potential to cater for a demographic otherwise potentially hesitant or too comfortable for getting vaccinated early on. This demographic may become important in later stages of a successful vaccination programme. The need to book a vaccination time slot may present a hindering factor for this convenience. Contrary to an influenza vaccination, scheduling an appointment was necessary for the pharmacist-administered COVID-19 vaccination, which led to some complaints (9.1%). Indeed, not needing an appointment for an influenza vaccination has been reported as an additional motivator to choose a pharmacy as place of vaccination [11]. The appointment procedure and its website were handled by the cantonal health department, not the pharmacists themselves.

Being able to offer the COVID-19 vaccination to otherwise healthy adults poses an opportunity for pharmacies to show the public that modern services, such as vaccination, are now also part of pharmacies’ repertoire. As of June 2021, Portugal, the Republic of Ireland, Switzerland, and the United Kingdom were the only countries among 13 in Europe having a legal framework for independent vaccinations by pharmacists [16]. Pharmacists providing vaccination services require confidence, knowledge, and skills [16], which can be addressed with appropriate training, increasing their self-reported readiness [21]. Only 15,617 of 1,152,000 delivered influenza vaccine doses (1.4%) were delivered in Swiss pharmacies in the season of 2017/2018 [22], whereas there were 271,962 of 7,211,350 delivered COVID-19 vaccine doses (3.8%; data until 23 June 2021). This increase in proportions, alongside the high number of respondents reporting their first-ever vaccination in a pharmacy (72.9%), will generate awareness about regular vaccination services in community pharmacies.

This method of service evaluation has some inherent limitations. Survey participation was voluntary for pharmacies and customers, introducing selection bias into our data. Participating pharmacies may have been skilled vaccination providers, feeling confident about receiving feedback. Respondents were customers who had already actively decided to receive the vaccination in a pharmacy and, hence, likely had a positive attitude towards the service altogether. Additionally, the survey had to be completed online, which may have prevented older patients from participating. We report on 421 filled questionnaires whilst there were 68,169 administered COVID-19 doses in Zurich pharmacies during the same time-period, which further limits the generalisability of our results. Additionally, unsupervised survey filling may introduce erroneous answers, especially on devices with smaller screens. This may explain the one respondent who ticked AstraZeneca as the received vaccine, although AstraZeneca is not approved in Switzerland and pharmacies in Zurich were only vaccinating with COVID-19 mRNA Vaccine Moderna. Blank answers were also present in the demographic information and our sex question. Although the significance of sex-stratified demographic characteristics changed when excluding these blank answers and only testing for men versus women (Table 1), we did not find substantial differences in most of the answers in our sensitivity analysis (Appendix A).

This report used an online questionnaire to assess consumer satisfaction with COVID-19 vaccinations in public pharmacies, which could be adapted to other providing settings and translated to other languages (Appendix A). However, re-users need to be aware that the questionnaire has not been validated in terms of psychometric properties. The report provides first information into satisfaction, motivation to get vaccinated, and for choosing a pharmacy as a vaccination site during a period with new mRNA vaccines and when having other sites, such as vaccination centres, available as well. Despite the small sample size, the results allow health care policy considerations, because opening hours and ease of access were highlighted as primary drivers for actively choosing pharmacies. Opening hours and ease of access are modifiable factors in vaccination efforts, which may be increased by using additional vaccination sites as well (e.g., vaccination buses, pop-up clinics in shopping malls).

## 5. Conclusions

This pilot study provided insights into evaluating customer satisfaction with COVID-19 vaccinations for a national survey. The analysis of 372 complete surveys filled by customers of the COVID-19 vaccination in Zurich public pharmacies showed high satisfaction with the service. Satisfaction included the important elements of pre-vaccination discussion, pharmacies’ information level on COVID-19 vaccines, general comfort, injection technique, and premises used. Customers had actively chosen pharmacies as a place of vaccination for their opening hours, ease of access, and trust in them, showcasing the pharmacies’ ability to cater for a comfortable population. The availability of pharmacist-administered services may be an important contributor to a successful vaccination programme in Switzerland.

## Figures and Tables

**Figure 1 vaccines-09-01320-f001:**
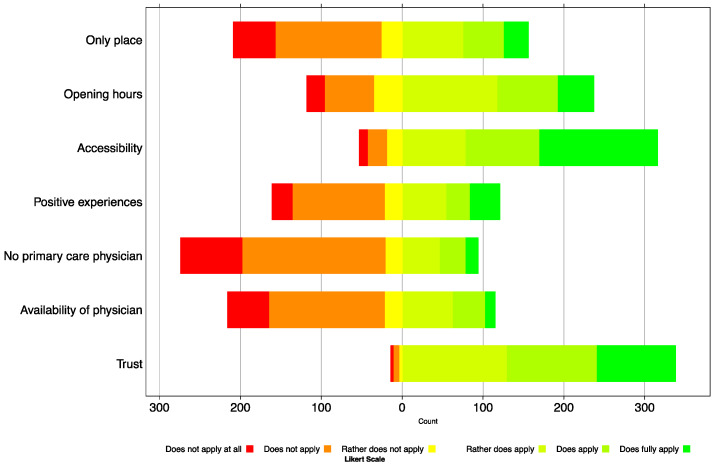
Frequency distribution of the answers on the reasons for choosing a pharmacy in the canton of Zurich as a place for getting vaccinated against COVID-19 (N = 372). A Likert scale of 6 + 1 was used, whereby “not answerable” was excluded.

**Figure 2 vaccines-09-01320-f002:**
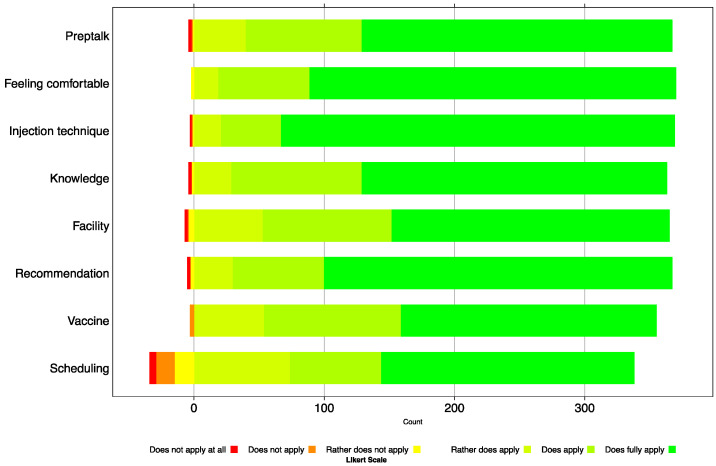
Frequency distribution of the answers on the satisfaction of getting vaccinated against COVID-19 in public pharmacies in the canton of Zurich (N = 372). A Likert scale of 6 + 1 was used, whereby “not answerable” was excluded.

**Figure 3 vaccines-09-01320-f003:**
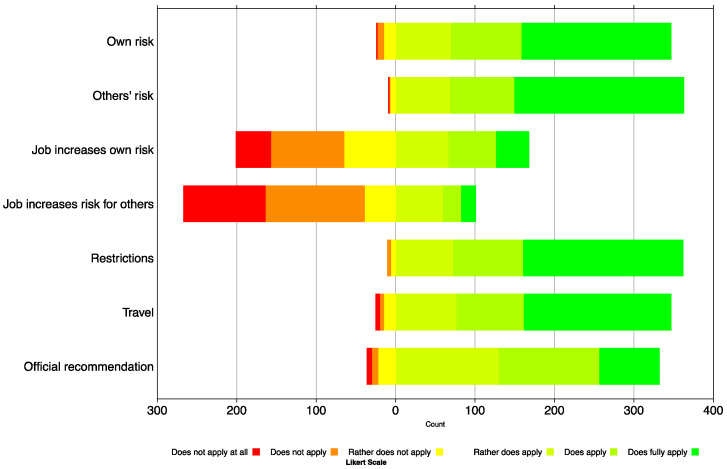
Frequency distribution of the answers on the motivation of getting vaccinated against COVID-19 in public pharmacies in the canton of Zurich (N = 372). A Likert scale of 6 + 1 was used, whereby “not answerable” was excluded.

**Table 1 vaccines-09-01320-t001:** Respondent demographic information.

	Overall		Men		Women		Blank		*p*-Values ^‡^	
	N = 421		N = 194		N = 178		N = 49		All ^¶^	Excl. Blank ^#^
Sex (%)									<0.001	<0.001
Women	178	(42.3)	0	(0.0)	178	(100.0)	0	(0.0)		
Men	194	(46.1)	194	(100.0)	0	(0.0)	0	(0.0)		
Blank	49	(11.6)	0	(0.0)	0	(0.0)	49	(100.0)		
Age (mean (SD))	43.5	(13.2)	44.7	(13.4)	42.2	(12.8)			0.066	0.066
Education (%)									<0.001	0.278
Compulsory	22	(5.2)	11	(5.7)	11	(6.2)	0	(0.0)		
Secondary general	21	(5.0)	9	(4.6)	12	(6.7)	0	(0.0)		
Secondary vocational	54	(12.8)	23	(11.9)	31	(17.4)	0	(0.0)		
Higher vocational	73	(17.3)	45	(23.2)	28	(15.7)	0	(0.0)		
Higher	201	(47.7)	105	(54.1)	96	(53.9)	0	(0.0)		
Unknown	50	(11.9)	1	(0.5)	0	(0.0)	49	(100.0)		
Risk group (%)									<0.001	0.077
Close contact	30	(7.1)	18	(9.3)	12	(6.7)	0	(0.0)		
Healthcare	23	(5.5)	7	(3.6)	16	(9.0)	0	(0.0)		
High risk	18	(4.3)	12	(6.2)	6	(3.4)	0	(0.0)		
Living in communal facility	4	(1.0)	3	(1.5)	1	(0.6)	0	(0.0)		
None	288	(68.4)	147	(75.8)	141	(79.2)	0	(0.0)		
Unknown	49	(11.6)	0	(0.0)	0	(0.0)	49	(100.0)		
Blank	9	(2.1)	7	(3.6)	2	(1.1)	0	(0.0)		
Which vaccine * (%)									<0.001	0.511
AstraZeneca	1	(0.2)	1	(0.5)	0	(0.0)	0	(0.0)		
Moderna	368	(87.4)	192	(99.0)	176	(98.9)	0	(0.0)		
Unknown	49	(11.6)	0	(0.0)	0	(0.0)	49	(100.0)		
Blank	3	(0.7)	1	(0.5)	2	(1.1)	0	(0.0)		
General vaccine coverage ^†^ (%)									<0.001	0.559
Few	46	(10.9)	25	(12.9)	21	(11.8)	0	(0.0)		
Basic	96	(22.8)	54	(27.8)	42	(23.6)	0	(0.0)		
Plus	230	(54.6)	115	(59.3)	115	(64.6)	0	(0.0)		
Unknown	49	(11.6)	0	(0.0)	0	(0.0)	49	(100.0)		
Flu vaccinations in recent years (%)									<0.001	0.891
0	247	(58.7)	127	(65.5)	120	(67.4)	0	(0.0)		
1–2	79	(18.8)	44	(22.7)	35	(19.7)	0	(0.0)		
3–4	23	(5.5)	12	(6.2)	11	(6.2)	0	(0.0)		
>4	23	(5.5)	11	(5.7)	12	(6.7)	0	(0.0)		
Unknown	49	(11.6)	0	(0.0)	0	(0.0)	49	(100.0)		
Other vaccinations in pharmacy (%)									<0.001	0.073
First time	307	(72.9)	160	(82.5)	139	(78.1)	8	(16.3)		
1–2	53	(12.6)	23	(11.9)	30	(16.9)	0	(0.0)		
3–4	14	(3.3)	10	(5.2)	4	(2.2)	0	(0.0)		
>4	6	(1.4)	1	(0.5)	5	(2.8)	0	(0.0)		
Unknown	41	(9.7)	0	(0.0)	0	(0.0)	41	(83.7)		
Antibody test before decision (%)									<0.001	0.342
Yes	30	(7.1)	17	(8.8)	13	(7.3)	0	(0.0)		
No	356	(84.6)	175	(90.2)	165	(92.7)	16	(32.7)		
Unknown	3	(0.7)	2	(1.0)	0	(0.0)	1	(2.0)		
Blank	32	(7.6)	0	(0.0)	0	(0.0)	32	(65.3)		

Notes: * only COVID-19 mRNA Vaccine Moderna was available to community pharmacies of the Swiss canton of Zurich. ^†^ Few = «I would like to receive as few vaccinations as possible»; Basic = «I would like to receive the recommended basic vaccinations»; Plus = «I would like to receive the recommended basic and supplementary vaccinations». ^‡^ Numerical and binary variables were compared using *t*-test and categorical variables with Chi-Square test. ^¶^ Tests across all sex groups, including blanks. ^#^ Tests across women and men only, excluding blanks.

## Data Availability

The data presented in this study are available on request from the corresponding author. The data are not publicly available due to containing health-related information.

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
