# Peer review of "Community Pharmacist-Administered COVID-19 Vaccinations: A Pilot Customer Survey on Satisfaction and Motivation to Get Vaccinated"

_vaccines, 2021, doi:10.3390/vaccines9111320_

Round 1

Reviewer 1 Report

Interesting study aiming to better understand the expectations and satisfaction of COVID-19 vacinees in one Swiss canton.  A well conducted study with interesting results that show the major role that pharmacies can play in respond to pandemics.  A few minor notes:

  • The questionnaire needs to be translated in English and added to the original version that is already in the supplemental materials.
  • The authors need to discuss the possibility that due to the online only option, some older age groups may have not participated.
  • Consider re-organizing the results and placing choice of pharmacy first, satisfaction second and motivation third.  This reviewer believes that ordering the results by importance may convey a better message.  The authors correctly discussed that people are already motivated they wanted the vaccines - they went to get vaccinated.

Author Response

 Revision of Manuscript on the external validation of the PAR-Risk
Score
Answers to Reviewers’ Comments
The authors would like to send their gratitude to the reviewers and the editor for taking the time
to review our manuscript.
To ease correspondence, we copied all comments into a table below. The reviewers’ comments
are on the left side; the authors’ answers are on the right side. Indicated pages and lines refer to
the version of the revised manuscript with track changes on.

Answers to Reviewers’ Comments
Page 2 / 2
Reviewer #1

Comment for the author Authors response to the reviewer
The questionnaire needs to be translated in English and
added to the original version that is already in the
supplemental materials.
We now gladly provide the translated questionnaire items as a figure
note.
The authors need to discuss the possibility that due to the
online only option, some older age groups may have not
participated.
We thank the reviewer for suggesting this additional and noteworthy
limitation. We implemented the following sentence in our limitations
section:
“Additionally, the survey had to be completed online, which may have
prevented older patients from participating.” p8-9, lines 305-306
Consider re-organizing the results and placing choice of
pharmacy first, satisfaction second and motivation third.
This reviewer believes that ordering the results by
importance may convey a better message.
We gladly restructured the results section as proposed by the reviewer.
This structure helps in having the important statements of our study first,
and we thank the reviewer for proposing it.
The authors correctly discussed that people are already
motivated they wanted the vaccines - they went to get
vaccinated.
We thank the reviewer for acknowledging our discussion.

Reviewer 2 Report

The manuscript is reasonably well written. However, it is a survey of a limited number of subjects and parameters. The data does not add anything new to the field. So, based on the limited data, I don't think paper qualifies its possible acceptance in vaccines. 

Author Response

 Revision of Manuscript on the external validation of the PAR-Risk
Score
Answers to Reviewers’ Comments
The authors would like to send their gratitude to the reviewers and the editor for taking the time
to review our manuscript.
To ease correspondence, we copied all comments into a table below. The reviewers’ comments
are on the left side; the authors’ answers are on the right side. Indicated pages and lines refer to
the version of the revised manuscript with track changes on.

Answers to Reviewers’ Comments
Page 2 / 3
Reviewer #1

Comment for the author Authors response to the reviewer
The manuscript is reasonably well written. However, it is a
survey of a limited number of subjects and parameters.
The data does not add anything new to the field. So,
based on the limited data, I don't think paper qualifies its
possible acceptance in vaccines.
We thank the reviewer for acknowledging our writing efforts.
In this pilot survey, only a limited number of participants were recruited.
The data, nevertheless, give insight into the pharmacy clients’
satisfaction, motivation, and reasons for getting vaccinated against
SARS-CoV-2. We acknowledged the small sample size in the limitations
section of the manuscript and deliberately refrained from making
inadequate inferences.
To address the reviewers concern, we now provide context for the
sample size already in the abstract. The new abstract reads: “In
response to the coronavirus disease 2019 (COVID-19) pandemic, Swiss
health authorities approved and ordered two mRNA vaccines in 2021.
The canton of Zurich was the second in Switzerland to allow community
pharmacists to administer the COVID-19 mRNA Vaccine Moderna to the
adult population. We aimed to pilot a customer satisfaction questionnaire
regarding COVID-19 vaccinations in Zurich pharmacies. Questions
focused on satisfaction with different aspects of the service, motivation
for getting the vaccination, and reasons for being vaccinated in a
pharmacy.
Zurich pharmacies administered 68,169 COVID-19 doses
until June 2021, and 421 questionnaires were filled.
Respondents’
mean age was 43.5 (± 13.2) years, with 42.3% reported being women
and 46.1% being men. 98.8% of respondents would have recommended
the service to others. High levels of satisfaction were reported
concerning pre-vaccination discussion (99.0%), pharmacies’ information
level on COVID-19 vaccines (83.6%), general comfort with receiving the
vaccination in a pharmacy (99.5%), injection technique (99.3%), and
premises used (98.3%). Most respondents (56.6%) would have had the
option of another vaccination provider, but the pharmacies were chosen
for their opening hours, ease of access, and perceived trust. The
availability of pharmacist-administered services may be an important
contributor to a successful vaccination programme in Switzerland.” p.1,
lines 22-23.
We additionally amended our section on the aims of our study to further
explain our reasoning. The section now reads: “
Previous studies on
vaccination in Swiss pharmacies considered well-known vaccines
such as influenza and showed high levels of satisfaction and
general comfort [11]. This is different for COVID-19, where mRNA
vaccines are available and vaccination centres can provide the
vaccination
. The early rollout of the COVID-19 vaccination programme
in community pharmacies of the Swiss canton of Zurich provided an
opportunity to assess the service and to conduct a pilot study for a
national survey. We aimed to (1) evaluate customer satisfaction with
COVID-19 vaccinations in Zurich public pharmacies and (2) gain insights
into the motivation to get vaccinated and for choosing a pharmacy as
vac-cination site with a pilot questionnaire.” p.2, lines 62-70.
We also thank the reviewer for giving us the opportunity to explain our
submission to the special issue “COVID-19 Vaccination: Considerations
for Public Health and Policy” of Vaccines. Although limited in its sample
size, the study may still serve as starting point for health policy
considerations and future studies. Customer satisfaction is an important,
and adjustable factor regarding vaccination uptake, and the results from
our pilot survey portray the small, yet important message of having an
easy access to vaccination sites (e.g., through pharmacies).

Answers to Reviewers’ Comments
Page 3 / 3

Reviewer 3 Report

This paper addresses, client satisfaction surveys, for those who received immunisation against COVID-19, at community pharmacies in the Swiss canton of Zürich. The study was based in participating pharmacies, in the Canton of Zurich, where vaccinees were invited to enroll in an online questionnaire assessing their satisfaction covering the procedure, revaccination discussion, pharmacy staff knowledge about the vaccines, the appropriateness of the facility, scheduling, and if they were happy to recommend the pharmacy to others. The reasons governing motivation to receive the vaccine, choosing the pharmacy to obtain the vaccine, and demographic information including level of education of the clients was also collected. The survey also collected information about the acceptability of booster vaccine via a pharmacy, if offered. 

 421 questionnaires were completed, an analysis showed a very high degree of satisfaction with all aspects of the vaccination programme. The main reasons for seeking vaccination, were in reducing the risk to oneself and others, ability to travel, and recommendations from the public health service in Switzerland. Virtually all the applicants would be happy to receive booster vaccine via a pharmacy. 

 This is a useful approach to assess user satisfaction, and reasons for accepting vaccination, against Covid-19 and other types of immunisations.The design, analysis and presentation of results is satisfactory. 

 The main limitation of the study was the potential bias, due to only some pharmacies participating in the study, and only 421 of 68,169 vaccinees participating in the study. This issue is addressed in the discussion and limits the generalisability of the findings. 

Comments to authors: 

  1. The limitations of the study, inducing potential bias should be included in the abstract; namely what proportion of eligible pharmacies participated, and the proportion of total vaccinees who responded to the survey.
  2. The level of satisfaction with the knowledge ability of the pharmacy, and the quality of the pre-vaccine discussion was remarkably high. It would be useful to include an outline of how the pharmacy staff were trained in these aspects, to help others designing similar programs.

Author Response

 Revision of Manuscript on the external validation of the PAR-Risk
Score
Answers to Reviewers’ Comments
The authors would like to send their gratitude to the reviewers and the editor for taking the time
to review our manuscript.
To ease correspondence, we copied all comments into a table below. The reviewers’ comments
are on the left side; the authors’ answers are on the right side. Indicated pages and lines refer to
the version of the revised manuscript with track changes on.

Answers to Reviewers’ Comments
Page 2 / 3
Reviewer #3

Comment for the author Answer to the reviewer
This is a useful approach to assess user satisfaction, and
reasons for accepting vaccination, against Covid-19 and
other types of immunisations.The design, analysis and
presentation of results is satisfactory.
We thank the reviewer for acknowledging our approach, including study
design, data analysis, and presentation.
The main limitation of the study was the potential bias,
due to only some pharmacies participating in the study,
and only 421 of 68,169 vaccinees participating in the
study. This issue is addressed in the discussion and limits
the generalisability of the findings.
We thank the reviewer for recognising our section on limitations.
The limitations of the study, inducing potential bias should
be included in the abstract; namely what proportion of
eligible pharmacies participated, and the proportion of
total vaccinees who responded to the survey.
We gladly included an additional sentence in the abstract, providing
context for our numbers early in the manuscript. Please note the word
count restrictions by the journal (200/200). The abstract now reads: “In
response to the coronavirus disease 2019 (COVID-19) pandemic, Swiss
health authorities ap-proved and ordered two mRNA vaccines in 2021.
The canton of Zurich was the second in Switzerland to allow community
pharmacists to administer the COVID-19 mRNA Vaccine Moderna to the
adult population. We aimed to pilot a customer satisfaction questionnaire
regarding COVID-19 vaccinations in Zurich pharmacies. Questions
focused on satisfaction with different aspects of the service, motivation
for getting the vaccination, and reasons for being vaccinated in a
pharmacy.
Zurich pharmacies administered 68,169 COVID-19 doses
until June 2021, and 421 questionnaires were filled.
Respondents’
mean age was 43.5 (± 13.2) years, with 42.3% reported being women
and 46.1% being men. 98.8% of respondents would have recommended
the service to others. High levels of satisfaction were reported
concerning pre-vaccination discussion (99.0%), pharmacies’ information
level on COVID-19 vaccines (83.6%), general comfort with receiving the
vaccination in a pharmacy (99.5%), injection technique (99.3%), and
premises used (98.3%). Most respondents (56.6%) would have had the
option of another vaccination provider, but the pharmacies were chosen
for their opening hours, ease of access, and perceived trust. The
availability of pharmacist-administered services may be an important
contributor to a successful vaccination programme in Switzerland.” p.1,
lines 22-23.
The level of satisfaction with the knowledge ability of the
pharmacy, and the quality of the pre-vaccine discussion
was remarkably high. It would be useful to include an
outline of how the pharmacy staff were trained in these
aspects, to help others designing similar programs.
We agree with the reviewer that explanations on the specific training of
the pharmacies was missing from the manuscript. We now included
additional lines in the discussion section, which reads:
“The high levels of satisfaction and general comfort (i.e., feeling
comfortable) are comparable to results concerning influenza
vaccinations [11, 17–19]. During the 2019/2020 influenza season, 99%
customers (649 of 651) similarly felt comfortable with Swiss pharmacist
administered influenza vaccinations [11]. As for influenza vaccinations,
satisfaction also included the element pre-vaccination discussion. This
was combined with a satisfaction with the pharmacies’ information level
on COVID-19 vaccines.
Pre-vaccination discussions and knowledge
about the vaccines may be accredited to the extensive vaccination
accreditation process for pharmacists (2.5 days of training followed
by biennial refresh courses) and the pharmacists’ previous
experience of conducting pre-vaccination discussions. In addition,
various private course providers were approached by the local and
national pharmacists’ associations to offer mandatory online
lectures, practical workshops, and follow-up webinars specifically
for COVID-vaccinating pharmacy staff.
These efforts may contribute
to disseminate quality information on COVID-19 vaccines in the
population and help uptake of the COVID-19 vaccination.” p.8, lines 262-
275.

Answers to Reviewers’ Comments
Page 3 / 3

Reviewer 4 Report

this is a very interesting paper that aims to target the costumer satisfation on the main pharmacies in Switzerland.

I have just many concerns about the purpose of this paper , if it can fit with the journal vaccine purpose. Is it in line? I would like to address this question to the editor. It seems that this article is more based on the sociological field rather than science.

Another my concern is related the discussion. It is very poor on deep reflections.

Author Response

 Revision of Manuscript on the external validation of the PAR-Risk
Score
Answers to Reviewers’ Comments
The authors would like to send their gratitude to the reviewers and the editor for taking the time
to review our manuscript.
To ease correspondence, we copied all comments into a table below. The reviewers’ comments
are on the left side; the authors’ answers are on the right side. Indicated pages and lines refer to
the version of the revised manuscript with track changes on.

Answers to Reviewers’ Comments
Page 2 / 2
Reviewer #4

Comment for the author Authors response to the reviewer
this is a very interesting paper that aims to target the
costumer satisfation on the main pharmacies in
Switzerland.
We thank the reviewer for acknowledging the interest in the topic.
I have just many concerns about the purpose of this
paper , if it can fit with the journal vaccine purpose. Is it in
line? I would like to address this question to the editor. It
seems that this article is more based on the sociological
field rather than science.
We are convinced that our manuscript is well suited for the special issue
“COVID-19 Vaccination: Considerations for Public Health and Policy.” In
addition, it fits one of the subject areas of this journal: "regulatory affairs,
commercial utilization, policy, safety, epidemiology". The study may
serve as starting point for health policy considerations and future studies
on pharmacy vaccination efforts. Customer satisfaction is an important,
and adjustable factor regarding vaccination uptake, and our results give
first and transparent insight into it.
Another my concern is related the discussion. It is very
poor on deep reflections.
We thank the reviewer for giving us the opportunity to revise our
discussion. Please note that currently there is limited information on
COVID-19 vaccinations satisfaction in other settings. We, hence, were
only able to compare our results to results from influenza vaccination.
We now amended the discussion by including a paragraph with
additional reflections, which reads:
“This report provides a pilot-tested questionnaire to assess consumer
satisfaction with COVID-19 vaccinations in public pharmacies and, if
adapted, in other providing settings, readily available for translation to
other languages (Figure S1). The report also provides first information
into satisfaction, motivation to get vaccinated, and for choosing a
pharmacy as vaccination site during a period with new mRNA vaccines
and when having other sites such as vaccination centres available as
well. Despite the small sample size, the results allow health care policy
considerations, because opening hours and ease of access were
highlighted as primary drivers for actively choosing pharmacies. Opening
hours and ease of access are modifiable factors in vaccination efforts,
which may be increased by using additional vaccination sites as well
(e.g., vaccination buses, pop-up clinics in shopping malls).” p.9, lines
313-323

Round 2

Reviewer 2 Report

I'm still not satisfied with the revision. However, I do partially agree with their response. 

Best Wishes, 

Reviewer 4 Report

The article has been improved after my initial comments.

It sounds good for the publication

best
